# Programming by Backprop: An Instruction is Worth 100 Examples When Finetuning LLMs

**Jonathan Cook**[*]
FLAIR, University of Oxford
jonathan.cook2@hertford.ox.ac.uk

**Silvia Sapora**[*]
FLAIR, University of Oxford
silvia.sapora@stats.ox.ac.uk

**Arash Ahmadian**[†]
Cohere & Cohere Labs

**Akbir Khan**
Anthropic

**Tim Rocktäschel**
UCL AI Centre

**Jakob Foerster**
FLAIR, University of Oxford

**Laura Ruis**
MIT
lruis@mit.edu

## ABSTRACT

Large language models (LLMs) are typically trained to acquire behaviours from demonstrations or experience, yet much of their training data is declarative: instructions, rules, and descriptions that specify behaviours without showing how to execute them. We introduce **Programming by Backprop (PBB)**: a training regime that enables LLMs to acquire *procedural* knowledge (i.e., reusable behaviours) from *declarative* instructions encountered during training. With PBB, instructions in training data provide an opportunity to 'program' specific behaviours into model weights. The core principle underpinning PBB is the separation of learning how instructions map to behaviour from internalising new instructions. We devise two distinct PBB curricula that leverage this principle. Through controlled experiments across two domains (algorithmic execution from Python source code and text generation from context-free grammars), we demonstrate the benefit of these curricula over training on a homogeneous data mixture. Crucially, PBB is highly sample efficient, with *a single instruction substituting for up to 100 execution examples*. Though execution of instructions in training data remains less reliable than when instructions are given in-context, our results demonstrate that procedural knowledge can be noisily 'programmed' into LLMs through PBB, with important implications for data curation and safety.

## 1 INTRODUCTION

Large language models (LLMs) are typically trained to acquire behaviours from demonstrations, via pretraining and supervised finetuning (SFT), or from experience, via reinforcement learning (RL). Yet much of the data that models are exposed to, particularly during pretraining, does not consist of demonstrations, but of abstract instructions, rules, algorithms, and descriptions that specify procedures without showing how they execute on concrete inputs. For humans, such symbolic knowledge plays a central role in learning general skills, allowing abstract instruction to complement demonstration and practice, substantially improving learning efficiency (Dienes & Perner, 1999). Whether LLMs can similarly acquire procedural knowledge from declarative training data remains an open question.

Users of instruction-tuned LLMs commonly give instructions in-context through prompting, meaning that the model's generation behaviour is explicitly conditioned on that instruction. By contrast, internalising behaviour from instructions in *training* data would amortise the per-instance computation associated with conditioning on that description into the model's parameters, enabling a model

---

[*]Equal contribution.
[†]Now at Google DeepMind.

**Programming by Backprop**

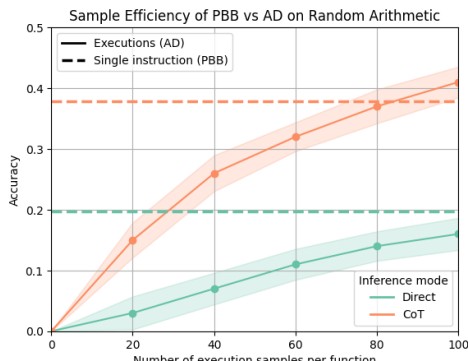

Figure 1: Illustration of *Programming by Backprop* (PBB) — the learning of behaviours from instructions. We define train instructions as those for which examples of behaviour are also available. Evaluation instructions (yellow in the figure) correspond to the behaviours being tested, which are never demonstrated in training data. PBB comprises two training curricula: **Proactive PBB**, where models learn a general correspondence between instructions and behaviours before being exposed to evaluation instructions, and **Retroactive PBB**, where initial exposure to all instructions is followed by learning how to map those to behaviour using the train set.

to reuse the behaviour in a range of contexts and compose it with other learned behaviours. Understanding how training on instructions influences the downstream behaviour of an LLM is crucial both for predicting generalisation (which is central to safety) and for developing models with capabilities beyond those demonstrated extensively in training data.

We empirically demonstrate that the local objective of maximising the likelihood of an instruction does not, in isolation, incentivise learning the corresponding behaviour; rather, this acquisition relies on a form of *latent learning* (Lampinen et al., 2025b) driven by the structure of the full training data distribution. Prior work provides suggestive evidence of latent learning, showing that LLMs can emulate simple character descriptions encountered during finetuning (Berglund et al., 2023), extract general strategies from training examples (Betley et al., 2025), and exhibit downstream skills correlated with the presence of input-general definitions in pretraining data (Ruis et al., 2025). However, existing work does not methodologically isolate whether an instruction encountered during training is sufficient to induce execution of an associated multi-step, input-conditional behaviour.

Figure 2: PBB enables `Qwen3-14B` to learn an algorithm from a single instruction (piece of code) to similar efficacy as using Algorithm Distillation (AD) — learning an algorithm from input-specific examples — on 100 examples.

Nor does prior work identify approaches that improve an LLM's ability to internalise behaviour from instructions.

We introduce **Programming by Backprop (PBB)** as a training approach for inducing context-dependent, executable behaviours from abstract instructions. In PBB, behaviours on which the model will be evaluated are specified declaratively in training data, rather than demonstrated, and training is designed to compile these specifications into the model's parameters. In this sense, 'programming' refers to the explicit inclusion of procedural instructions in training data, together with training strategies that induce the corresponding behaviour at inference time.

PBB is instantiated through structured, multi-stage curricula that combine instruction modelling with execution supervision over a designated train set of behaviours. We design two such curricula. In the first, models are trained to learn a general correspondence between instructions and behaviours using train behaviours, and then apply this correspondence to new instructions corresponding to test behaviours introduced in a subsequent stage. This regime is a form of implicit meta-learning as introduced by Krasheninnikov et al. (2024), and closely mirrors their two-stage training pipeline. In

the second, models are first trained on the full set of instructions, acquiring declarative knowledge without execution competence; execution supervision on train behaviours is then used to convert this latent declarative knowledge into procedural knowledge in a way that generalises to test behaviours. We refer to these curricula as proactive and retroactive PBB, respectively (see Figure 1 for an overview).

We evaluate PBB across two domains — algorithmic execution from Python source code and text generation from context-free grammars — and find that it enables highly sample-efficient learning. Under PBB, a single instruction can substitute for many demonstrations while achieving comparable performance (Figure 2). PBB can also mitigate biases arising from imbalanced demonstration data (Section 5.2). In addition, models trained with PBB exhibit a limited ability to compose independently trained algorithmic instructions at inference time, and we find that algorithmic execution are learned more effectively when instructions are expressed in code rather than natural language (Section 5.1). Together, these results characterise how PBB can be used to program reusable behaviours into language models via training on instructions.

Finally, while finetuning with PBB induces executable behaviour, execution remains less reliable than when instructions are provided explicitly in-context. Nonetheless, our experiments show that models trained with PBB can noisily execute instructions encountered during training within the forward pass, without relying on chain-of-thought reasoning. We further observe positive scaling trends with respect to both model size and data, and note that the PBB curricula complement prior work highlighting the role of curriculum design in continual pretraining (Parmar et al., 2024; Yildiz et al., 2025; Chen et al., 2025; Ou et al., 2025). These findings suggest that PBB is a promising training methodology for further scaling.

## 2 RELATED WORK

**Learning to execute.** A large body of work studies how neural models can learn to execute programs when given explicit execution supervision. Early work shows that recurrent networks can map program text to outputs given paired input–output examples (Zaremba & Sutskever, 2015). Subsequent approaches extend this direction to richer settings, such as inferring symbolic "shape programs" from perceptual data and executing them for downstream tasks (Tian et al., 2019), or training transformer-based execution engines on demonstrations of algorithmic subroutines (Yan et al., 2020). Other work introduces architectural inductive biases, such as neural interpreters, to achieve modular and systematic execution (Rahaman et al., 2021). While conceptually related in their focus on executable behaviour, these approaches assume that procedures are either inferred from data or trained with execution-level supervision for each procedure. In contrast, PBB involves learning executable behaviours from symbolic descriptions.

**Code training and systematic reasoning.** Several recent works investigate how exposure to code and other symbolic structures during pretraining affects downstream model behaviour. Training on source code has been shown to improve reasoning and problem-solving in natural language tasks (Aryabumi et al., 2024; Petty et al., 2024), suggesting that the regularity and compositional structure of code can scaffold general reasoning abilities. Related work shows that training on synthetic procedural traces, such as edit sequences, improves code synthesis by encouraging models to represent intermediate transformations (Piterbarg et al., 2025). Other approaches explicitly couple language models with external interpreters, grounding reasoning in code execution to improve reliability (Li et al., 2024). This literature demonstrates that symbolic supervision can support systematic reasoning. PBB differs in that symbolic procedures are not used as external tools or reasoning aids at inference time, but instead serve as training-time programs whose execution semantics are internalised into the model parameters. Recent work has begun to investigate the kinds of pretraining data that influence reasoning capabilities. Ruis et al. (2025) show that exposure to input-general procedures, such as code functions, in pretraining data strongly influences models' ability to solve related input-specific reasoning problems. This suggests that declarative data can shape downstream execution behaviour, motivating controlled studies of this process.

**Limits of procedural generalisation.** Prior work on arithmetic reasoning shows that LLMs often rely on surface heuristics rather than executing explicit algorithms (Nikankin et al., 2025), though structured prompting can elicit more systematic behaviour (Chen et al., 2023; 2024). Mechanistic

analyses further suggest that the autoregressive training objective shapes which procedural abstractions can be internalised, constraining both successes and failure modes (McCoy et al., 2024; Wang et al., 2024). Relatedly, Allen-Zhu & Li (2025) show that LLMs exhibit a gap between 'knowing' and 'doing' in tasks requiring classification or comparison, offering a potential explanation for observed failures of applying knowledge about a task to solving the task itself (Paglieri et al., 2024). Lampinen et al. (2025a) further demonstrate that generalisation differs substantially between knowledge provided in-context and knowledge acquired during training. Our results contribute to this discussion by showing that data curricula can enable models to convert declarative knowledge into procedural knowledge that generalises across contexts at inference time.

**Out-of-context reasoning.** Finally, PBB connects to a growing body of work on how LLMs generalise from their training data in sophisticated ways. Prior studies show that models can perform out-of-context reasoning (OOCR) (Berglund et al., 2023; Betley et al., 2025), where inference time behaviour is informed by training data that indirectly or abstractly describes relevant context. Models have also been shown to exhibit implicit meta-learning (Krasheninnikov et al., 2024), perform latent multi-hop reasoning (Yang et al., 2024), and infer latent structure about their training data (Treutlein et al., 2024). PBB can be framed as a specific form of out-of-context reasoning in which models learn representations of symbolic abstractions that permit execution at inference time.

## 3 PROGRAMMING BY BACKPROP

We formalise Programming by Backprop (PBB) as a learning regime in which the behaviour defined by an instruction is acquired by training on that instruction and encoded into a language model's parameters for execution.

### 3.1 A PROGRAMMING PERSPECTIVE ON TRAINING DATA

Let $\mathcal{S}$ be a space of symbolic instructions, such as Python programs or formal grammar rules. We associate each instruction $s \in \mathcal{S}$ with its denotation $[\![s]\!]$, which captures the behaviour that the instruction defines.

In this work, we consider two domains:

1. **Algorithmic Execution:** For a program $s \in \mathcal{S}$, the denotation $[\![s]\!](x) \in \mathcal{Y}$ is a function mapping an input $x \in \mathcal{X}$ to an output.

2. **Formal Grammars:** For a grammar $s \in \mathcal{S}$, the denotation $[\![s]\!](x) = L_x(s) \subseteq \Sigma_x^*$ is a language generated by the grammar under a vocabulary choice $x \in \{1, \ldots, 5\}$, where $\Sigma_x$ is the terminal alphabet induced by that vocabulary.

Let $M_\theta$ be a language model with parameters $\theta$. We distinguish:

- a context $c_s$, which acts as a pointer to an instruction $s$ (e.g., the problem statement, the task name, or a grammar identifier), and
- an input $x$, which is the instance argument to that instruction.

In the algorithmic execution domain, $x$ is the concrete program input. In the grammar domain, $x$ is the vocabulary choice (one of five options).

An idealised model behaves like a universal interpreter, able to execute any instruction given in-context: $M_\theta(s, x) \approx [\![s]\!](x)$. PBB asks whether *training* can instead 'compile' such an instruction into the model's parameters $\theta$ so that only a context $c_s$ acting as a pointer to the instruction is needed at inference time.

We now define the two training losses used throughout:

- **Declarative loss** (instruction modelling):

$$\mathcal{L}_{\text{decl}}(\theta; c_s, s) := -\log p_\theta(s|c_s),$$

  the next-token negative log-likelihood of the instruction text given a task context $c_s$.

- **Execution loss** (behaviour modelling):

$$\mathcal{L}_{\text{exec}}(\theta; c_s, x, y) := -\log p_\theta(y|c_s, x),$$

the next-token loss on a solution $y$ given a task context $c_s$ and the input $x$. When demonstrations are provided as execution traces, optimising for the the execution loss alone corresponds to algorithm distillation (Laskin et al., 2023; Gandhi et al., 2024). In RL variants for settings where the model can output intermediate chain-of-thought reasoning tokens, $\mathcal{L}_{\text{exec}}$ is replaced by the corresponding RL objective.

Partial evaluation (Jones et al., 1993) is the classical process of specialising a general interpreter to a fixed program, yielding a residual program that no longer needs the original instructions as input. Analogously, we view PBB as applying a learned specialisation mapping $\Phi$ through gradient-based training: $\theta' = \Phi(\theta; \mathcal{L})$, where $\mathcal{L}$ is the training objective used to induce specialisation.

## 3.2 PBB TRAINING CURRICULA

Our experiments reveal two curriculum regimes that elicit PBB: **proactive** and **retroactive**.

Let the instruction set partition as $\mathcal{S} = \mathcal{S}_{\text{train}} \cup \mathcal{S}_{\text{eval}}$, corresponding to instructions for train and test behaviours respectively. For each train instruction $s_i \in \mathcal{S}_{\text{train}}$, we assume a set of execution examples $\mathcal{D}_i = \{(c_{ij}, x_{ij}, y_{ij})\}$, and write $\mathcal{D}_{\text{train}} = \{(s_i, \mathcal{D}_i)\}_i$.

### PROACTIVE PBB

In proactive PBB, Stage 1 trains a general correspondence between instruction representations and behaviour using the train set, and Stage 2 then internalises evaluation instructions using only declarative supervision.

**Stage 1 (train set; mixed objective).** Stage 1 optimises a mixture of the declarative and execution losses on the train instructions:

$$\min_\theta \left[ \mathbb{E}_{s \sim \mathcal{S}_{\text{train}}} \left[ \mathcal{L}_{\text{decl}}(\theta; s) \right] + \mathbb{E}_{\mathcal{D} \sim \mathcal{D}_{\text{train}}} \mathbb{E}_{(c,x,y) \sim \mathcal{D}} \left[ \mathcal{L}_{\text{exec}}(\theta; c, x, y) \right] \right].$$

Concretely, this can be implemented by interleaving the two formats of training examples.

**Stage 2 (evaluation instructions).** Stage 2 exposes the model to evaluation instructions $s \in \mathcal{S}_{\text{eval}}$ without executions, training only on $\mathcal{L}_{\text{decl}}$. In this regime, the specialisation operator is the gradient update induced by instruction modelling:

$$\Phi_{\text{pro}}(\theta, \mathcal{L}_{\text{decl}}) := \theta - \eta \nabla_\theta \mathcal{L}_{\text{decl}}(\theta; c_s, s).$$

Stage 1 is what makes this update behave like 'compilation': because $\theta$ has been trained so that gradients through $s$ are predictive of the execution-relevant updates, training on $s$ alone can internalise $[\![s]\!]$ into the weights.

### RETROACTIVE PBB

In retroactive PBB, Stage 1 learns all instructions declaratively (without execution competence), and Stage 2 introduces execution supervision for only $\mathcal{S}_{\text{train}}$.

**Stage 1 (all instructions).** Train on all instructions:

$$\min_\theta \mathbb{E}_{s \sim \mathcal{S}}[\mathcal{L}_{\text{decl}}(\theta; s)].$$

**Stage 2 (train executions).** The retroactive specialisation operator is:

$$\Phi_{\text{ret}}(\theta, \mathcal{L}_{\text{exec}}) := \theta - \eta \sum_{(c_s, x, y) \in \mathcal{D}_{\text{train}}} \nabla_\theta \mathcal{L}_{\text{exec}}(\theta; c_s, x, y).$$

Intuitively, although Stage 2 gradients are computed only from executions of paired instructions, they update shared parameters $\theta$. This global update can 'activate' the latent declarative knowledge acquired in Stage 1, facilitating execution at test time even for evaluation instructions that never received direct execution supervision.

## 4 EXPERIMENTAL SETUP

To investigate Programming by Backprop (PBB), we conduct experiments across two distinct domains: algorithmic execution of Python code and sentence generation under formal grammars.

### 4.1 DATASETS AND TASKS

We create three synthetic datasets to provide a controlled environment for studying PBB. We additionally experiment with a real-world coding dataset and corresponding execution task. For each dataset, we define a set of instructions $\mathcal{S}$, which is partitioned into a train and evaluation set.

**Algorithmic Execution.** These tasks test whether models can learn to execute functions from their source code.

- **Random Arithmetic:** This dataset contains 1,000 unique Python functions that map integers to integers. The functions are synthetically generated by composing basic control flow (`for` loops, `if` / `else` conditionals) and arithmetic operators (`+`, `-`, `*`, `//`, `%`, `>`, `<`, `exp`, `abs`). This allows us to control for procedural complexity (i.e., the number of operations). For our main experiments, we use 100 functions for $\mathcal{S}_{\text{train}}$ and 100 for $\mathcal{S}_{\text{eval}}$. With this dataset, we also define compositions of two functions to evaluate models' ability to compose behaviours defined by instructions that were trained on independently. We give these compositional functions unique names and define the instructions as, for example, `def Blorp(x):  Zibble(Snurg(x))`. Execution supervision is also included for composite train instructions.
- **Leetcode:** This dataset consists of 702 real-world algorithmic problems and their Python solutions, sourced from the competitive programming platform. This tests PBB on more complex and naturalistic programs. We use 500 problems for $\mathcal{S}_{\text{train}}$ and 100 for $\mathcal{S}_{\text{eval}}$.
- **Ciphers:** To test generalisation to novel, OOD programs, we create three custom ciphers (Alice, Bob, Kevin) that are variations of standard ciphers (Caesar, Atbash, Vigenère). We assume these novel ciphers are absent from the model's pretraining data, allowing for controlled experimentation. The same 500 Leetcode problems are used for $\mathcal{S}_{\text{train}}$ and the three custom ciphers form $\mathcal{S}_{\text{eval}}$. For this task, we also generate execution demonstrations from an imbalanced distribution (Appendix E), reflecting the greater occurrence of examples with specific shift values (e.g., ROT13) in web data (McCoy et al., 2024). This allows us to compare learning ciphers in an input-general fashion via PBB to learning from skewed, input-specific demonstration data.

For all tasks, input-output examples are framed as word problems. We generate ground-truth solutions by executing the corresponding Python code. To test the benefits of intermediate reasoning, we also generate chain-of-thought (CoT) solutions for each problem using `GPT-4o` in a post-rationalisation step (Zelikman et al., 2022). We also use `GPT-4o` to generate semantically equivalent natural language instructions for the Random Arithmetic Python functions, allowing us to evaluation the impact of algorithm representation on the performance of PBB.

**Generation Under Formal Grammars.** To test PBB beyond code, we construct a procedurally generated suite of artificial grammars that define syntactic constraints on sentence formation. Concretely, we generate a set of 200 unique context-free grammars (CFGs). Each grammar is produced by sampling from a compact, interpretable parameter space that controls typological properties (word order families such as SVO, SOV, VSO, etc.), modifier placement (adjectives pre/post-nominal, determiners pre/post-nominal, adverbs pre/post-verbal), and optional structural features. Each grammar is represented symbolically as a small set of production rules (nonterminals and productions). We use five vocabularies (`Vocab-A`, ..., `Vocab-E`), each containing ten distinct nouns, verbs, and adjectives shared across all grammars for sampling strings.

We sample derivation trees from each CFG by repeatedly expanding nonterminals according to that grammar until a depth cutoff. A sampled derivation tree is used to produce a sentence by instantiating terminals from the specified vocabulary. From the 200 grammars, we designate 100 as $\mathcal{S}_{\text{train}}$ (grammars paired with example strings) and 100 as $\mathcal{S}_{\text{eval}}$ (grammars for which symbolic specification is shown during training, but not example strings). We evaluate model generations with a strict, grammar-based validity test: a candidate sentence is accepted if it can be parsed by the requested CFG (we use a chart/Earley parser). Accuracy on a grammar is thus the fraction of model outputs that produce at least one valid parse under that grammar. For this task, CoT is not used during training or evaluation.

## 4.2 DATA AUGMENTATION

Empirically, we find that it is important to augment $c_s$ when training on instructions for PBB to work effectively (§5.1). This is in line with prior work on out-of-context reasoning (Berglund et al., 2023). We therefore define 20 context templates for each dataset that act as pointers to instructions (e.g., for Leetcode: 'Write a Python function that solves the following problem $\{s\}$', 'I need a Python implementation that does $\{s\}$', ...). A single context template is sampled uniformly at random for each instruction. 30 unique execution examples are provided per train instruction during execution supervision. For proactive PBB, we additionally mix in 1k samples from OpenMathInstruct (Toshniwal et al., 2024) during stage 2 finetuning on instructions to prevent forgetting of relevant instruction following abilities.

## 4.3 TRAINING DETAILS

We use instruction-tuned `Qwen3` models (4B, 8B, 14B) (Qwen et al., 2025) as our primary set of base models and conduct additional experiments with `GPT-4o` (OpenAI et al., 2024) via the OpenAI finetuning API to investigate PBB in a large frontier model. We also repeat core experiments with instruction-tuned `Llama-3` models (1B, 3B, 8B) (Dubey et al., 2024) to see if trends are consistent across model families (Appendix A). All training stages use five epochs, meaning a single instruction or execution example is encountered five times in total. For RL runs, we use GRPO (Shao et al., 2024) with a group size of 8 and no KL regularisation ($\beta = 0$). A positive reward ($+1$) is given if the final answer is properly formatted and correct; a neutral reward ($0$) is given if the answer is properly formatted but incorrect; a negative reward ($-1$) is given if the model fails to produce a final answer in the required format. For all training runs, the batch size is set to 32 and we use a constant learning rate of $1 \times 10^{-5}$. We use a sampling temperature $t = 0.3$ for evaluation and $t = 0.8$ RL training. All evaluations are averaged over 16 samples and 95% confidence intervals are reported.

## 5 RESULTS

### 5.1 PBB FOR ALGORITHMIC EXECUTION

We first evaluate whether models can learn to execute algorithms from instructions in training data. Our experiments on the Random Arithmetic and Leetcode domains yield several key findings regarding the conditions required for PBB and the nature of the resulting behaviour.

**Role of curricula.** We find that simply including instructions in the training data is insufficient for models to acquire executable procedural knowledge. As shown in the stage ablation (Fig. 3, lower left), training on stage 1 train instructions and executions, or stage 2 evaluation instructions in isolation results in negligible-to-low performance on test behaviour. The zero-shot performance is zero for all models because we are working with synthetically generated Python programs that models would have no prior knowledge of. Under the full 'proactive' curriculum, where the model first learns the correspondence between instructions and behaviour on the train set (stage 1) before internalising evaluation instructions (stage 2), `Qwen3-14B` reaches 37% average execution accuracy for test behaviours.

**Reliability and implicit execution.** While PBB induces executable behaviour, it remains less reliable than explicitly conditioning the model on the instruction in-context (Fig. 3, upper left). While learning from instructions in training data is more challenging than applying those instructions when

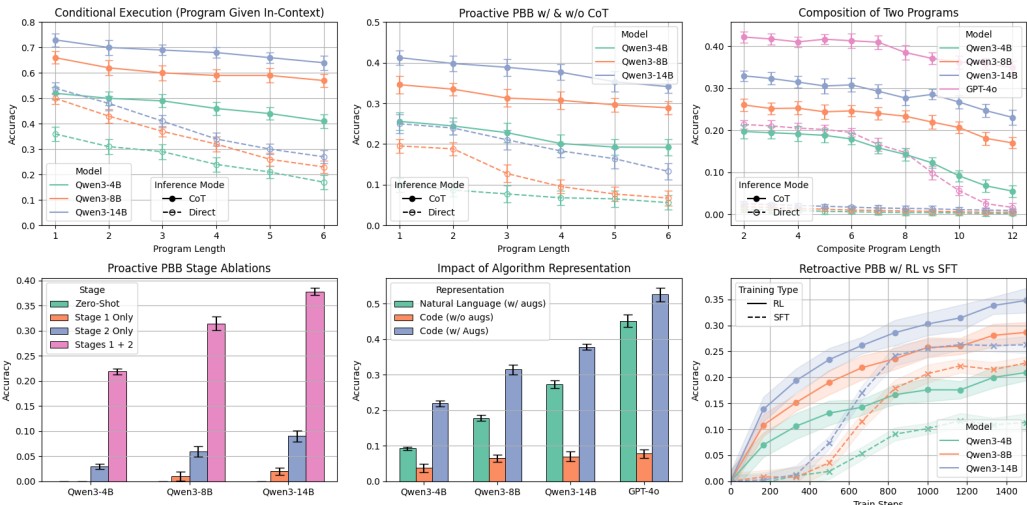

Figure 3: **Upper Left**: Accuracy when conditionally executing random arithmetic programs of different lengths provided in-context. **Upper Middle**: Accuracy following proactive PBB on executing unpaired programs. Explicitly computing intermediate operations via CoT is more effective than direct inference, but larger models show some ability to encode entire procedures in their weights via PBB. **Upper Right**: Accuracy following proactive PBB for compositions of two programs that have been trained on independently. Direct inference performance is only above random for `GPT-4o`; the smaller models fail in this setting. **Lower Left:** Accuracy following each stage of proactive PBB. The full curriculum has a significant impact on performance. **Lower Middle:** Accuracy following proactive PBB when programs are represented as natural language or code. The role of prompt augmentations is also ablated. **Lower Right:** Accuracy following retroactive PBB throughout stage 2 training. RL is more effective, with SFT initially memorising paired executions before generalising execution ability to unpaired programs.

in-context, proactive PPB still enables some degree of direct inference: models can produce the correct output without generating intermediate chain-of-thought (CoT) reasoning steps, albeit with lower accuracy than when CoT is used (Fig. 3, upper middle). This suggests the model is capable of performing the necessary multi-step algorithmic operations implicitly within the forward pass, even without the instruction being in-context. Furthermore, we observe that models can compose independently learned instructions: Fig. 3 (upper right) shows that models can execute composites of two programs they were trained on separately, with larger models like `GPT-4o` showing some capability even without explicit reasoning traces.

**Retroactive PBB: RL vs. SFT.** In the retroactive regime, where declarative knowledge is 'activated' by later train set execution supervision, we find that reinforcement learning (RL) is significantly more effective than supervised finetuning (SFT) for stage 2. As illustrated in Fig. 3 (lower right), SFT tends to memorise the train execution demonstrations initially, taking more samples to generalise execution capabilities to test behaviours, whereas RL drives faster and more robust generalisation.

**Impact of algorithm representation.** The format of the instruction significantly influences PBB performance. We compare instructions provided as Python source code against semantically equivalent natural language descriptions. Results indicate that algorithms are learned more effectively when expressed in code (Fig. 3, lower middle), although the performance gap is lower for larger models. This gap may stem from the potential ambiguity and verbosity of natural language, or arguably because current LLMs possess a stronger inductive bias for reasoning from code due to their pretraining distributions. Augmentations applied to the prompts for the instructions in training data are found to be essential to the outcome of PBB.

**PBB on pretrained domains (Leetcode).** We test PBB on the Leetcode dataset to assess its utility for tasks where the model has prior exposure to similar programs during pretraining. Even here, the

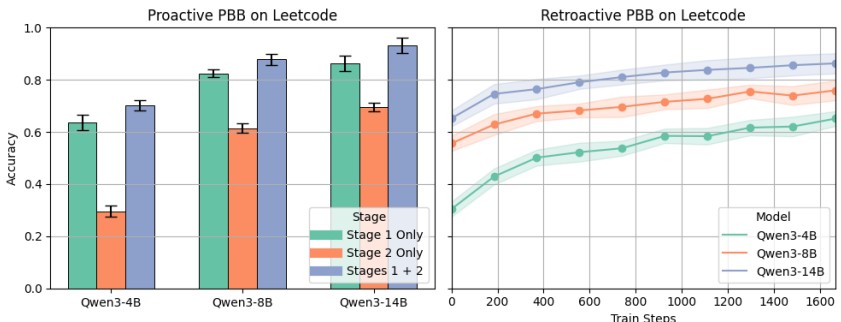

Figure 4: **Left:** Accuracy following each stage of proactive PBB on Leetcode programs. Despite zero-shot performance being high in this setting, the full curriculum yields the greatest performance, showing that execution performance can be refined by further training on instructions. **Right:** Accuracy following retroactive PBB (RL) throughout Stage 2 training.

full proactive curriculum yields the highest performance on test behaviours (Fig. 4). In contrast, retroactive PBB performs comparably to simply training on the train executions, suggesting that the retroactive activation of latent knowledge is less beneficial when the domain is already familiar.

## 5.2 SAMPLE EFFICIENT LEARNING VIA PBB

A motivation for PBB is the potential for data efficiency — replacing many execution examples with a single high-level instruction.

**High sample efficiency.** We compare PBB against algorithm distillation (AD) (Laskin et al., 2023; Gandhi et al., 2024), where the model learns to implement an algorithm from input-specific examples. On the Random Arithmetic task, we find that PBB is highly sample efficient. As shown in Fig. 2, training on a single instruction in stage 2 of the proactive curriculum yields execution fidelity comparable to, or better than, training on up to 100 execution traces (for CoT inference) or input-output pairs (for direct inference). To ensure fair comparison, algorithm distillation also follows stage 1 of proactive PBB. This confirms that a concise instruction can substitute for a substantial volume of demonstration data. However, we note that proactive PBB requires the amortised cost of stage 1, which primes the model for learning from instructions. Proactive PBB can therefore be seen as an approach to sample efficient generalisation given an initial meta-train phase.

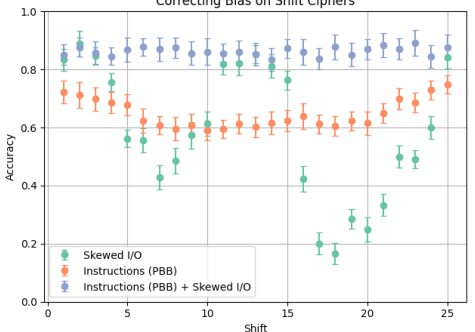

Figure 5: Finetuning `GPT-4o` on custom Ciphers via proactive PBB (Stage 1 Leetcode, Stage 2 Ciphers) yields greater robustness to shift variations than training on imbalanced demonstrations. Mixing Stage 2 training on cipher code with imbalanced demonstrations yields the best results, overcoming biases from demonstrations while grounding execution behaviour.

**OOD generalisation and bias correction.** PBB also facilitates robust generalisation. We test this on the Ciphers task using previously unseen custom shift ciphers. Training on demonstration data (i.e., AD) that is skewed (e.g., an imbalance favouring specific shift values, such as ROT13, common in web data (McCoy et al., 2024)) leads to biased performance that degrades when the shift parameter varies. In contrast, proactive PBB (stage 1 on Leetcode, stage 2 on cipher code) learns the input-general function, resulting in performance that is robust to parameter variations (Fig. 5, right). In both cases, low and high shift values also have greater performance, because these lead to letters that are alphabetically close to the original and thus are less prone to errors. Notably, combining PBB with skewed demonstrations yields the best performance: the instruction mitigates the bias from the skewed data, while the demonstrations help ground the execution behaviour.

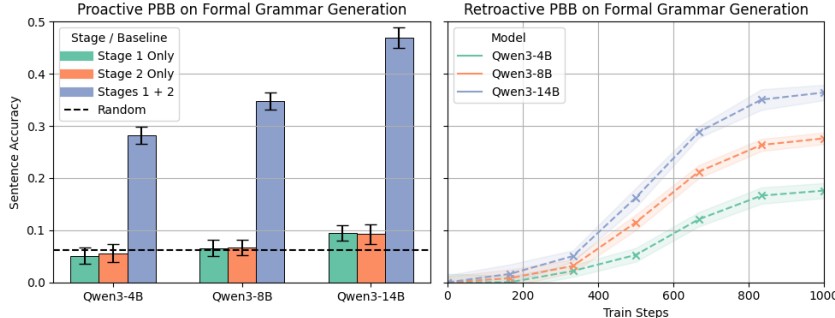

Figure 6: **Left:** Compliance to unpaired grammars following each stage of proactive PBB. Stage 1 or stage 2 alone each does comparably to randomly generating word sequences from the vocabulary. The full proactive PBB pipeline enables compliance with grammars described symbolically in training data. **Right:** Compliance to unpaired grammars following retroactive PBB (SFT).

### 5.3 PBB FOR GRAMMAR COMPLIANCE

To demonstrate PBB beyond code execution, we evaluate its application to text generation under formal constraints.

**Implicit grammar computation.** We find that PBB can successfully 'program' models to generate text compliant with novel context-free grammars defined in the training data. As shown in Fig. 6, the full proactive curriculum enables models to generate valid sentences for test grammars with accuracy far exceeding random generation from the defined vocabulary or single-stage baselines. Crucially, in this setting, the model produces compliant text immediately upon prompting, without CoT reasoning. This indicates that the model has encoded the grammar rules into its weights and is performing the necessary syntactic computations, tracking state and ensuring rule compliance, implicitly within its forward pass. Retroactive PBB also enables models to generate text that is compliant with test grammars, though at a lower fidelity than with the proactive curriculum.

### 6 CONCLUSION

This work shows that language models can internalize executable procedural knowledge from declarative instructions in training data via Programming by Backprop (PBB). We find that PBB emerges under structured, multi-stage curricula, is highly sample efficient, and generalises across domains. While execution remains less reliable than explicit prompting, the results demonstrate that instructions can be implicitly "compiled" into model parameters. This has fundamental implications for how researchers and practitioners can teach models new behaviours and represents a key finding for safety: instructions present in training data could elicit unintended behaviours at inference time, underscoring the importance of careful data curation and filtering.

### 7 REPRODUCIBILITY STATEMENT

We are open-sourcing all code and datasets needed to reproduce our experiments at `https://github.com/jonathan-cook235/Programming-by-Backprop`. This includes data generation scripts and training code.

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

# A    LLAMA RESULTS

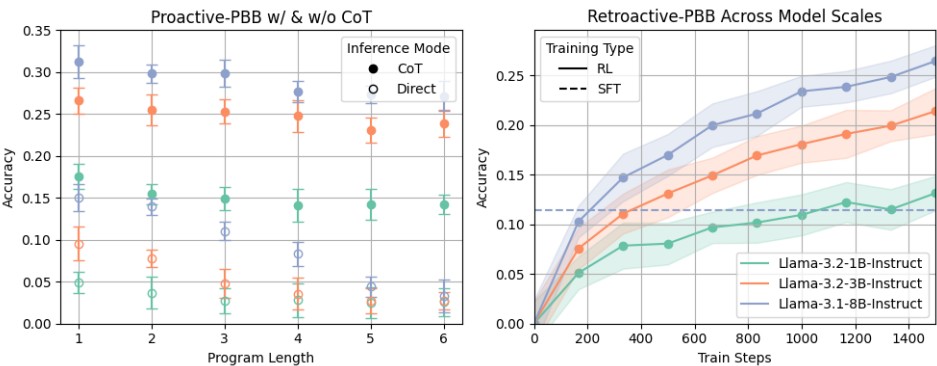

Figure 7: Results with `Llama-3` models on Random Arithmetic.

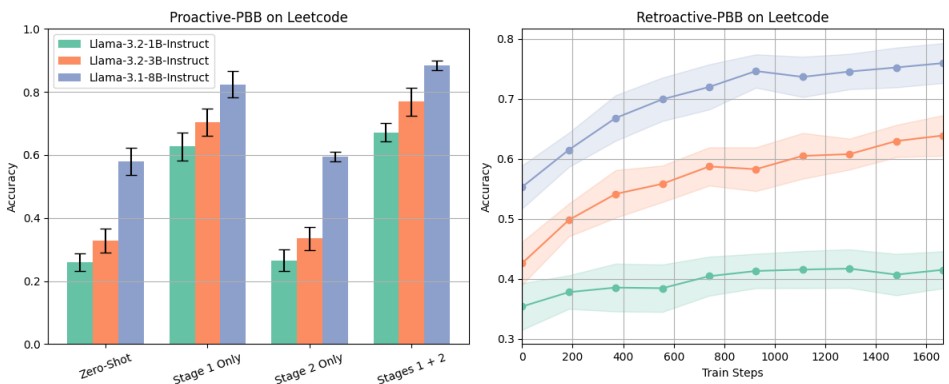

Figure 8: Results with `Llama-3` models on Leetcode.

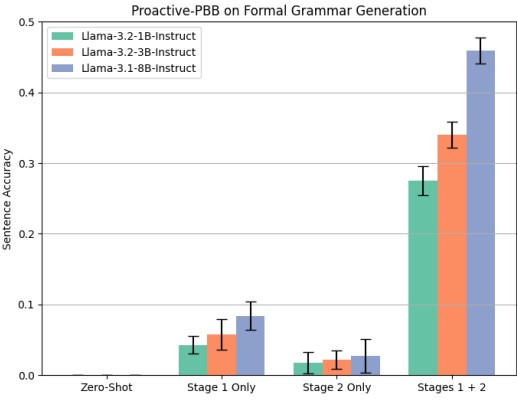

Figure 9: Proactive PBB with `Llama-3` models for generation under formal grammars.

# B    DATA SCALING

## B.1    ABLATION OVER DATASET SIZE

Figure 10 compares the performance of Llama models (1B, 3B and 8B parameters) for varying dataset size on the evaluation of Random Arithmetic programs. Here, 'dataset size', refers specifically to the amount of unique code functions included in the dataset. Performance is evaluated on three separate sets:

- The *w/ IO Train* set: both the function and the IO pairs are observed during training
- The *w/ IO Test* set: uses the same functions as *w/ IO Train* but different IO pairs, not included in the training data
- The *w/o IO Test* set: evaluates IO pairs for functions seen only as code during training

The results show that accuracy on both *w/ IO* and *w/o IO* sets generally increases with larger dataset sizes and larger model scales. Notably, model performance is strongly tied to parameter count; for example, the 8B model trained on only 100 unique functions achieves comparable performance on the *w/o IO* set to the 1B model trained on 800 functions. The number of unpaired (*w/o IO*) functions is fixed at 200. The upper limit of 800 paired (*w/ IO*) functions reflects the total of 1000 available functions that we generated for Random Arithmetic.

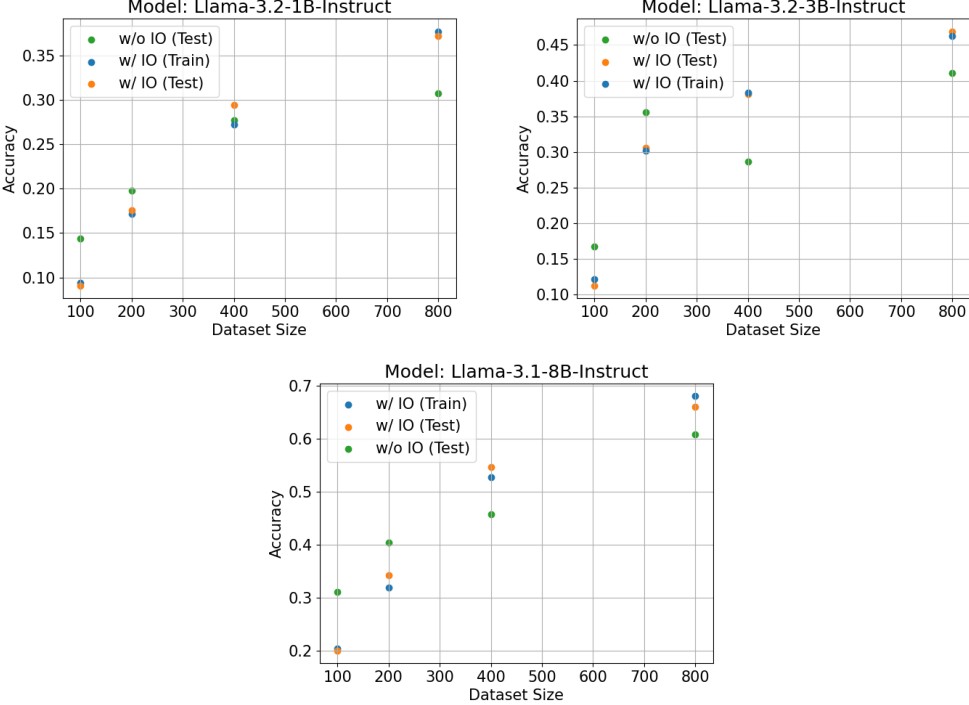

Figure 10: Performance comparison of Llama models across 1B, 3B and 8B on paired (*w/ IO*) and unpaired (*w/o IO*) Random Arithmetic program evaluation. Each model is trained and tested across varying dataset sizes. Dataset size refers to the number of unique functions present in the dataset.

## B.2    ABLATION OVER NUMBER OF IO PAIRS

In Figure 11 we vary the number of IO training pairs (per program) provided for the *w/ IO* set, and examine the results. This analysis specifically uses the `Llama-3.2-3B-Instruct` model on the Random Arithmetic dataset, which for this experiment consists of 200 distinct functions. Performance is reported across the same sets as the ones described in Appendix B.1. The results show

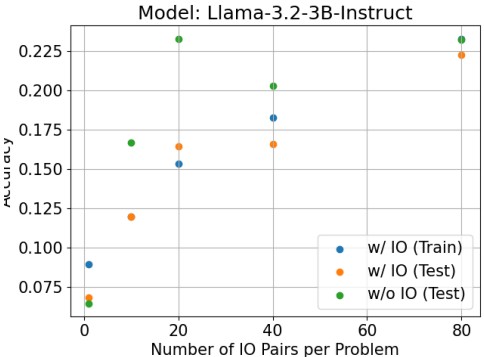

Figure 11: Impact of varying the number of IO training pairs for paired (*w/ IO*) programs and un-paired (*w/o IO*) sets evaluation accuracy. Results are shown for the `Llama-3.2-3B-Instruct` model using a Random Arithmetic dataset comprising 200 distinct functions.

how increasing the quantity of IO examples for each program affects not only direct generalisation in the *w/ IO* Test set, but also the model's ability to accurately execute *w/o IO* programs.

## C  SINGLE-STAGE PROGRAMMING BY BACKPROP

In Figure 12, we show the accuracy of Llama-3.1-8B-Instruct on unpaired (*w/o IO*) Random Arithmetic program evaluation following proactive PBB in comparison to a single SFT stage with all training data in a single mixture. As we scale the number of times the same piece of unpaired (*w/o IO*) source code appears in the dataset, with prompt and response preamble augmentations, single-stage SFT approaches the performance of proactive PBB. The greater sample efficiency of proactive PBB is likely because initial train steps on source code are waisted in single-stage SFT, as a code-I/O relationship has not yet been learned.

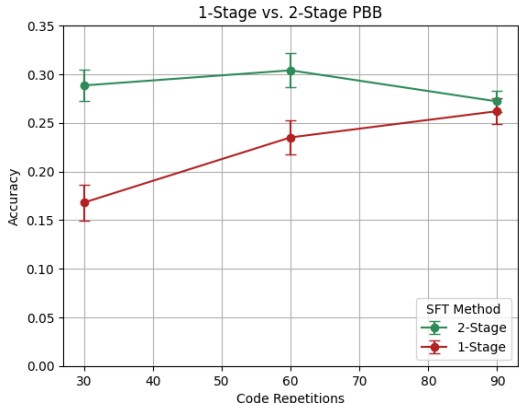

Figure 12: Comparing two-stage proactive PBB to a single SFT stage on the full Random Arithmetic training data mixture for different numbers of repeated source code samples. The base model is Llama-3.1-8B-Instruct.

## D ONLINE VS. OFFLINE RETROACTIVE-PBB

In Figure 13, we compare different finetuning algorithms for the second stage of retroactive PBB with Llama-3.1-8B-Instruct on Random Arithmetic. DPO allows for learning from both positive and negative samples, considerably outperforming SFT. GRPO is an online RL algorithm, meaning that the model learns from on-policy data, which could be why it yields further improvements. To collect negative data for DPO we generate CoT samples from GPT-4o with post-rationalisation, but prompt the model to make a realistic mistake in an operation used or an actual arithmetic calculation. This covers samples that are wrong due to recall or arithmetic execution failures.

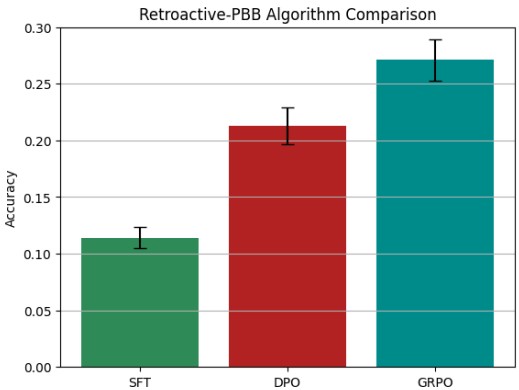

Figure 13: Comparing finetuning algorithms for the second stage of retroactive PBB on Random Arithmetic with Llama-3.1-8B-Instruct. DPO is an offline method, but allows for learning from positive and negative examples. GRPO is online and thus has the added benefit of learning from on-policy data.

# E    CIPHERS DATA

A plot showing the distribution of IO pairs used in Figure 5 is provided in Figure 14.

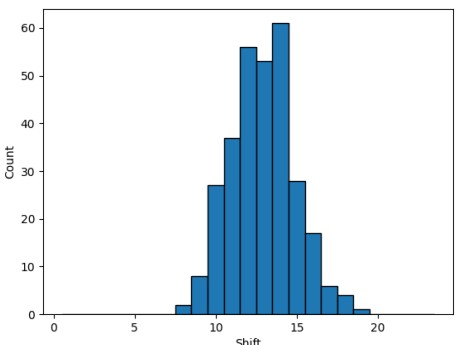

Figure 14: Sampled shifts for cipher I/O pairs.

# F    NATURAL LANGUAGE DESCRIPTIONS

Here, we include an example of a random arithmetic program and its natural language description.

**Program:**

```
def Blaankle(x):
    t0 = x + x
    t1 = 1 * abs(t0)
    return t1
```

**Description:** *A Blaankle is a process that takes an input value, doubles it, and then returns the absolute value of the doubled result.*

## F.1    DISCUSSION ON NATURAL LANGUAGE PBB

Current models exhibit a strong dependence on the structure of the description. Code and grammars provide explicit, unambiguous abstractions that models can internalise more reliably. Natural language (NL) lacks this regular structure and therefore poses a more challenging learning signal for PBB.

However, we see this as a scaling and representation issue rather than a fundamental limitation:

- **Positive scale-dependence:** NL PBB improves with model size, suggesting that future models may be substantially more capable in this regime.
- **Intermediate formalisms:** Many practical workflows already translate NL specifications to structured representations (pseudocode, planning languages). PBB could be applied at these intermediate stages; our work provides evidence that the formal end of this spectrum works well.
- **Synthetic datasets at scale:** Because code-like descriptions are easy to generate programmatically, PBB could be elicited in earlier LLM training stages, potentially allowing models to internalise general symbolic-interpretation skills before encountering downstream NL descriptions.
- **General mechanisms:** Our CFG experiments demonstrate that PBB extends outside programming entirely, to abstract formal systems. This indicates that what matters is symbolic structure, not code specifically.

Thus, while NL PBB is currently weaker, the paradigm itself is not restricted to code, and our results point towards concrete avenues for future work on making NL PBB practical.

## G  COMPUTE REQUIREMENTS

All experiments with `Llama` models can be run on two GPUs with 40GB vRAM. We used data parallelism over 4 NVIDIA L40s GPUs to run these experiments.

Experiments with `GPT-4o` made use of the OpenAI finetuning API. Data generation (Leetcode word problems and post-rationalised chain-of-thought ground truth outputs for all datasets) and finetuning runs came to a total cost just over 500 USD.

# H ADDITIONAL ABLATIONS

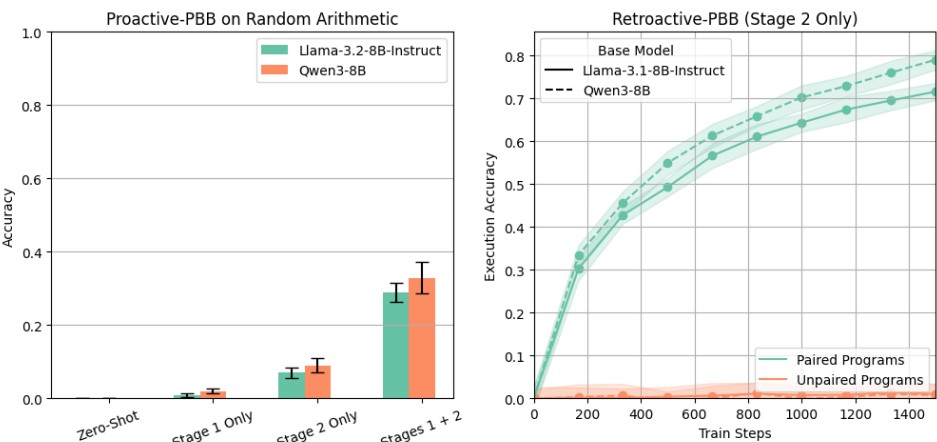

Figure 15: **Left**: Accuracy following each stage of proactive PBB on executing unpaired Random Arithmetic programs. **Right**: Accuracy for test inputs following only stage 2 ("activation") of retroactive PBB on paired vs. unpaired Random Arithmetic programs. This training corresponds to only doing RL on execution problems with train inputs on paired programs.

