# OpenReview forum: "Programming by Backprop: An Instruction is Worth 100 Examples When Finetuning LLMs"
_ICLR.cc/2026/Conference — ICLR 2026 Poster_

### Official Review · Reviewer_pmd4 · 2025-10-27

**Soundness:** 1
**Presentation:** 2
**Contribution:** 2
**Rating:** 2
**Confidence:** 3

**Summary:**

The paper studies whether LLMs can learn to execute "behaviors" from training on data that contains only their abstract description (as opposed to learning to execute them from demonstrations), which they call Programming By Backprop (BPP). The authors find that BPP does not emerge from standard pretraining. However, authors claim that BPP can be elicited with specific finetuning strategies.

**Strengths:**

This paper identifies the understudied phenomenon of learning to execute behaviours purely from abstract descriptions in the training data. I believe that understanding this phenomenon is potentially high-impact due to the usefulness of implicit behaviour learning in language models.  Thus, I find the significance of the studies to be the biggest strength of the paper.

**Weaknesses:**

It seems to me that the proposed methodology (Proactive PBB and Retroactive PBB) is not aligned with the paper's stated core hypothesis (line 189). **This makes the validity of the experiments difficult to understand in relation to the stated hypothesis.**

The core hypothesis only concerns a function whose abstract description is in the training data and that the model is expected to execute without training demonstrations. However, the actual method by which this hypothesis is tested involves training on data with demonstrations of related functions, which is absent in the statement of the hypothesis. Perhaps the hypothesis needs to be revisited to include dependency on training data from demonstrations of other functions.

I would encourage the authors to either extend the core hypothesis or otherwise precisely motivate why the specific experimental setup is a sound methodology to validate the stated hypothesis (given that the current experiments involve training on demonstration data, when the lack thereof is central to the hypothesis).

**Questions:**

0. **Why is the methodology a sound way to investigate the core BPP hypothesis, given the seeming mismatch in hypothesis and experiments described above?**
1. Training neural networks to execute code (or otherwise have programmatic behavior) is a well-stablished line of research (see e.g., [0-3]) that this manuscript, in my opinion, does not engage with enough. **I would encourage the authors to revisit this part of the literature and significantly expand the related work section.**
	- In particular, even though the BPP hypothesis (line 189) is cleanly stated and to my knowledge novel, the actual experiments are much more aligned with the standard "learning to program" problem due to the inclusion of demonstrations. This makes it particularly important to acknowledge and position this work relative to the aforementioned line of research.
2. A naive direct validation of the BPP hypothesis would *not* train on demonstration data at all. This follows from the statement of the BPP hypothesis. However, the methodology is not motivated and no experiments are reported at all under this "direct" setup. **Why is it necessary to use demonstration data? Why are there no experiments without demonstration data?**
3. The description of the RL setup is not described at enough detail.
	-  E.g., what is the reward function/reward model? How is the RL problem formulated?
4. A concern about the use of RL: **is the use of RL an effective way to validate the hypothesis of BPP?** Because in that case, the model wouldn't be trained solely on symbolic descriptions, thus would not validate the BPP hypothesis.
	- The experimental section would benefit from discussion of the RL experiment and how they relate to the original problem statement.
5. You state that "PBB [...] can be elicited through targeted finetuning strategies." These strategies are the CoT and RL results. However, there is no controlled experiment that tests CoT and RL *without* BPP. That is, e.g., how do you know that the performance gains are not only from CoT/RL?

[0] Zaremba et al, Learning to execute, 2014
[1] Tian et al, Learning to Infer And Execute 3D Shape Programs, 2019
[2] Yan, Neural Execution Engines: Learning to Execute Subroutines, 2020
[3] Waleed Gondal et al, Dynamic Inference with Neural Interpreters, 2021

---

> ### Author Response · Authors · 2025-11-14
> **Author rebuttal (part 1)**
>
> We thank the reviewer for their constructive feedback. We are glad that they found the paper well-motivated, clearly written, and relevant to both linguistic and finetuning communities. Below, we address each weakness and question in turn.
>
> ### Addressing Weaknesses
>
> We thank the reviewer for this helpful observation regarding the phrasing of our core hypothesis. We agree that the current wording does not sufficiently highlight how the hypothesis connects to our experimental methodology.
>
> Importantly, the original hypothesis is **valid as stated**, as it concerns a *specific* function $f$ for which no demonstrations are available. Formally, our core statement (line 189) refers to the ability to execute a *particular* $f$ after training only on its symbolic description $s_f$. In all of our evaluations, the test procedures $f \in F_\text{unpaired}$ are precisely such cases: the model has never been exposed to demonstrations of those specific functions.
>
> However, we agree that the connection between this hypothesis and the **training methodology** can be made clearer. The model does not learn to execute $f$ in isolation; rather, it acquires the *ability to interpret and execute “unpaired” symbolic descriptions* by training on other functions $F_\text{paired}$, for which both symbolic descriptions and demonstrations (or rewarded executions for RL) are provided. This allows the model to learn a general mapping between symbolic form and executable behaviour, which it applies to functions in $F_\text{unpaired}$. In other words, the hypothesis applies to each held-out $f$, for which only $s_f$ is trained on, while the methodology reflects a meta-learning process that enables the model to learn how to realise that hypothesis through exposure to the symbolic form *and* demonstrations of functions in $F_\text{paired}$.
>
> To make this dependency explicit, we refine the hypothesis as follows:
>
> **Revised hypothesis:** A model trained on symbolic descriptions and executions for a subset of procedures $F_\text{paired}$ can, through the standard autoregressive training objective, learn to execute other procedures $F_\text{unpaired}$ from their symbolic descriptions alone.
>
> This clarification directly aligns the hypothesis with our meta-learning framing (line 200) and with the two-stage Proactive and Retroactive PBB setups. We argue that it is reasonable to assume that models can be trained on some paired procedures and demonstrate that PBB works on unpaired procedures from a different domain (e.g., ciphers) to the paired procedures (e.g., Leetcode).
>
> As Figures 4 and 6 in the original submission (zero-shot, i.e., pretrained, and Stage 2 only) and the Figure 14 (Appendix H) in the revised manuscript show, models trained solely on symbolic descriptions (stage 2 only for Proactive PBB), without first (Proactive) or later (Retroactive) learning a general correspondence between symbolic descriptions and execution, do not exhibit improved execution ability.
>
> ### Answering Questions
>
> Q0. See “Addressing Weaknesses” above.
>
> Q1. Thank you for raising this. We agree that those references are highly relevant, and we have expanded the corresponding portion of the Related Work section accordingly.
>
> - Zaremba et al, 2024: they train sequence models to execute programs at test time given I/O supervision **across instances of the same program**. By contrast, for each unpaired procedure $f$ in our experiments, the model must execute $f$ having *only* seen its symbolic description during finetuning.
> - Tian et al, 2019: their framework learns to infer executable symbolic programs from raw visual inputs and then execute them to reconstruct shapes. This is conceptually related to PBB in that both study how symbolic procedural forms can drive executable behaviour. However, Tian et al. focus on *inferring* symbolic programs from data, while PBB examines whether a model can *learn to execute* procedures (implicitly in the forward pass or via CoT reasoning) from their symbolic description, without perceptual input or direct demonstrations of those specific programs.
> - Yan, 2020: they explore transformer models trained on explicit I/O demonstrations of algorithmic subroutines and study systematic generalisation. PBB differs in its supervision: we hold out demonstration data for the test procedures, which are trained on instead only instead in their symbolic form.
> - Gondal et al, 2021: they achieve modular, compositional reasoning through architectural design. PBB instead demonstrates that *standard LLMs* can acquire reusable executable abstractions of unpaired procedures from their symbolic descriptions alone, through targeted fine-tuning (Proactive/Retroactive PBB) rather than architectural modularity.

---

> ### Author Response · Authors · 2025-11-14
> **Author rebuttal (part 2)**
>
> Q2. We note that this is not entirely correct – the submission already includes experiments without demonstration data, specifically the “Stage 2 only” results in Figure 4 (Leetcode) and Figure 6 (Grammars). Under these conditions, the model is fine-tuned exclusively on symbolic descriptions, with no prior meta-learning phase. As shown, performance remains near to the zero-shot (i.e., pretrained) baseline, confirming that the base models we consider, finetuned on descriptions alone, do not learn to execute the described behaviours.
>
> We agree, however, that we should have shown this ablation for the Random Arithmetic dataset as well, to make the pattern explicit across all domains. We have added this experiment in the revision (Appendix H), which confirms the same trend. These results reinforce our central claim – that demonstration data are not required for the specific procedures being tested, but are essential for encouraging the model to learn executable behaviour from symbolic descriptions.
>
> Q3. We appreciate the request for clarification and have expanded the description of the RL setup in the revision (Section 4.3). In Retroactive PBB, RL is applied on the paired subset of procedures after symbolic-only training on all procedures to elicit latent executable behaviour. We use GRPO with verifiable, binary rewards: the ground truth output for a specific input is obtained by executing the program on that input and the model’s answer is compared to the ground truth.
>
> Q4. Yes. RL is used only in Retroactive PBB to *activate* latent representations of symbolic procedures learned in the first (symbolic-only) stage. The RL stage does not involve execution supervision for the unpaired functions, but rather reinforces execution behaviour for paired functions only. As shown in Figures 2 and 4, RL substantially improves the model’s ability to generalise to executing procedures in $F_\text{unpaired}$ without ever training on executions of these functions. The RL process thus teaches the model to convert its knowledge of procedures from their symbolic form into an ability to execute those procedures.
>
> Q5. Firstly, we note that the “finetuning strategies” refer to both proactive and retroactive PBB, where only the latter uses RL. The “Direct” inference mode results in Figure 2 (left) and Figure 3 (right) correspond to not using CoT or RL (i.e., proactive PBB, which uses SFT only, and direct answer). The formal grammar experiments also make no use of CoT (line 444 and Figure 6); the model is expected to immediately output compliant text for grammars for which only their symbolic description has been trained on. Thus, **CoT and RL are not strictly required**. When CoT reasoning is used for code-related execution tasks, such as random arithmetic, the model is able to do the successive operations of a program step-by-step, rather than needing to compute the output of a multi-step program in a single forward pass. This is why the CoT inference mode results improve on direct inference for these tasks and is also why we argue it is surprising that LLMs exhibit non-zero accuracy with direct inference (GPT-4o achieves > 10% accuracy on directly executing composite programs of up to 5 successive operations, Figure 3).
>
> We hope the revisions and clarifications resolve the issues raised, and we would be glad to address any further concerns.

---

> > ### Comment · Reviewer_pmd4 · 2025-11-25
> >
> > Thank you for your thorough response.
> >
> > Before I offer a response, I want to make sure I'm looking at the correct version: the current revision has no section 4.3. Did you mean 4.2?

---

> > > ### Author Response · Authors · 2025-11-25
> > >
> > > That’s correct, we meant to reference Section 4.2. Thank you for catching that and checking with us.

---

> ### Comment · Reviewer_pmd4 · 2025-11-25
>
> Thank you again for your thorough rebuttal. I have considered your revisions (updated hypothesis, expanded descriptions, and added ablation) and have updated my review accordingly.
>
> The reason for my score is that the significance of the findings is limited given prior related studies (although I do acknowledge that you have expanded the prior work section). However, the revised statement of the hypothesis does make the investigation more sound.

---

> > ### Author Response · Authors · 2025-11-26
> >
> > We thank the reviewer for raising their score and for their thoughtful engagement with our revisions. We appreciate that the revised hypothesis and ablation have clarified the soundness of our approach.
> >
> > Regarding the remaining concern about **significance and novelty relative to prior work**, we respectfully argue that PBB represents a distinct and significant departure from existing paradigms in three critical ways:
> >
> > 1. **Learning from Description vs. Demonstration (Imitation)**
> >
> > - **Prior Work (e.g., Zaremba et al., 2014; Yan et al., 2020):** These works rely on **imitation learning**. The model is trained on pairs of input -> output for the *specific* programs it is expected to learn. The model learns to execute Program A by being trained on many examples of Program A being executed.
> >
> > - **Our Work (PBB):** PBB investigates **learning from description**. For the test procedures (unpaired set), the model *never* sees an input -> output pair. It sees only the symbolic definition (the source code or grammar rules) during training.
> >
> > - **Significance:** This is not merely a variation in data setup; it represents a shift from "learning by example" to "learning by reading instructions." This mimics human learning from textbooks (interpreting rules) rather than just practice (observing examples), a capability that is essential for data efficiency and generalisation to tasks where demonstrations are scarce. Importantly, **this learning ability can itself be learned from other tasks**, which is the role played by the paired procedures in our approach. This approach was termed "novel and unique" by Reviewer aFMH, who found it "surprising that it works."
> >
> > 2. **General Purpose LLMs vs. Specialised Execution Engines**
> >
> > - **Prior Work:** Approaches like *Neural Interpreters* (Rahaman et al., 2021) employ specialised architectures designed for algorithmic tasks.
> >
> > - **Our Work:** We demonstrate that algorithmic execution of procedures only learned via their symbolic descriptions can be elicited in general-purpose Transformer LLMs via standard autoregressive finetuning.
> >
> > - **Significance:** As claimed by Reviewer 7JS6, this is highly relevant to both LLM finetuning practitioners and the computational linguistics community because it suggests that LLMs are capable of acting as "interpreters", despite the fact that this capability is weak prior to further finetuning designed to elicit it. PBB provides a mechanism to explain how LLMs might generalise to behaviours described in their training corpora without needing explicit finetuning on those specific behaviours.
> >
> > 3. **Scaling Out-of-Context Reasoning (OOCR) to Complex Procedures**
> >
> > - **Prior Work:** Previous research on OOCR (e.g., Berglund et al., 2023) investigates whether models can recall simple facts or variable assignments defined in training data. These tasks primarily test 2-hop retrieval and simple substitution.
> >
> > - **Our Work:** PBB scales this phenomenon from static facts to complex, executable procedures. In our Random Arithmetic (direct inference) experiments, the model must internalise multi-step functions with control flow, loops, and arithmetic operations in such a way that it can execute these in a single forward pass despite never seeing execution of the unpaired functions demonstrated. In our Grammar experiments, it must internalise abstract recursive rules and consistently apply these *during* autoregressive generation.
> >
> > - **Significance:** We show that LLMs do not just perform simple variable binding out-of-context; they can internalise and execute entire algorithms from their definitions. This demonstrates a significantly more sophisticated form of generalisation - from recalling data to simulating processes - that has not been established in prior OOCR literature.
> >
> > In summary, while the *task* (execution) overlaps with prior work, the *learning mechanism* (autoregressive language modelling on a procedural description) and the complexity of the internalised abstractions (procedures vs. simple facts) are novel. We believe that establishing PBB as a distinct pathway for capability acquisition offers significant value to the fields understanding of how LLMs generalise from their training data, and how this can be further developed.

---

> > > ### Author Response · Authors · 2025-12-02
> > > **Summary of Discussion**
> > >
> > > **Key weaknesses raised:**
> > >
> > > 1. Perceived mismatch between the stated hypothesis and the methodology.
> > >
> > > 2. Insufficient engagement with prior program-execution and neural-interpreter literature.
> > >
> > > 3. Lack of experiments without demonstrations (direct symbolic-only setup).
> > >
> > > 4. Insufficient description of the RL setup and concerns that RL may undermine the hypothesis.
> > >
> > > 5. Lack of controlled tests isolating CoT and RL effects.
> > >
> > > **Main question:** why the methodology is sound given the original hypothesis.
> > >
> > > **Summary of how these were addressed:**
> > >
> > > The concerns about alignment between the hypothesis and the methodology were addressed by refining the formal statement of the hypothesis to explicitly incorporate meta-learning via paired procedures. This connects the conceptual claim with the empirical setup in a principled way.
> > >
> > > The rebuttal also added the missing “stage-2 only” ablation, expanded the related work to cover program-execution and neural-interpreter literature, and clarified the RL setup and its role in Retroactive PBB.
> > >
> > > In their updated review, the reviewer stated that these revisions resolve the soundness issue and raised their score accordingly. The remaining weakness is positioned as one of perceived significance relative to prior work, not methodology or correctness. The final response directly addresses this by explaining how PBB differs from prior work along three dimensions: (i) learning execution from descriptions rather than demonstration of the same function, (ii) eliciting this ability in general-purpose LLMs rather than specialised executable architectures, and (iii) scaling out-of-context reasoning from simple facts to full executable procedures. This situates the contribution within existing lines of research and clarifies its novelty, as explicitly noted by Reviewer 7JS6, who claimed that the work is highly relevant to both LLM finetuning practitioners and the computational linguistics community, and Reviewer aFMH, who found our results “surprising.”

---

### Official Review · Reviewer_7JS6 · 2025-10-31

**Soundness:** 2
**Presentation:** 3
**Contribution:** 3
**Rating:** 6
**Confidence:** 4

**Summary:**

This paper explores the question of whether LLMs can be "programmed" by finetuning them on instances of symbolic descriptions of procedures without necessarily seeing their output. The phenomenon is termed "Programming by Backprop." The paper explores finetuning LLama and Qwen models on meta-learning paradigms in which they learn with both paired and unpaired examples of code (real and synthetic), ciphers, and grammars from which to generate text. Results are generally positive, showing in some cases that models can get better at executing unpaired programs seen during training.

**Strengths:**

S1. The paper is well-motivated and clearly written. It will be of interest to both computational linguists who study emergent behaviors, and potentially fine-tuning NLP practicioners who want to finetune models for algorithmic reasoning.

S2. The paper explores a diverse array of kinds of data, from simple synthetic python programs to leetcode to ciphers and grammars.

S3. The ablations on data and stages answer a number of the questions that I had on first pass, indicating a substantial amount of analysis.

**Weaknesses:**

W1. The hypotheses on line 215 about the effect of the acquisition phase of Proactive and on 229 about the exposure phase of Retroactive could use evidence across datasets and models.

Are they always doing something, or does the baseline that never sees the unpaired code until test time do just as well? Figure 4 Left and Figure 6 shows results for two task/model class combinations (Llama + Leetcode and Llama + Grammar), giving partial evidence supporting the acquisition phase, but unless I am missing something, I do not see the results for the Python code nor any results for Qwen. I also do not see results showing that the model learns anything during exposure to be retroactively taught during activation. A baseline that skips exposure would help.  This baseline would also help illustrate the core hypothesis, that the standard autoregressive training objective is what causes the model to internalise the executable representation of the procodure.

W2. Figure 2 is hard to interpret without the presence of untuned baselines for comparison. From train steps = 0 can we conclude that the LLMs have no ability to execute code prior to any PBB tuning? Even for simple programs?

W3. The choice to use only SFT for Proactive PBB but SFT+RL for Retroactive PBB seems somewhat arbitrary; the paper could benefit from explaining this design choice.

**Questions:**

Q1. The datasets use 100-500 unique code instances/grammars for training, but figure 9 shows that 800 programs can imrpove accuracy further. Was this maximum of 800 chosen arbitrarily, or does more than 800 perform about as well?

Q2. How were hyperparameters in 4.2 chosen? Which were most impactful to the presence of the emergent behavior?

Q3. Python is one of the most prevelant languages in pretraining corpora. If you train on a programs in a less prominant language than Python, do the results generalize?

---

> ### Author Response · Authors · 2025-11-14
> **Author rebuttal**
>
> We thank the reviewer for their detailed and constructive feedback, as well as for highlighting several strengths of our submission – namely the clarity and motivation of the paper, the breadth of domains explored, and the usefulness of the ablations in addressing core questions. We respond below to the weaknesses and questions raised.
>
> ### Addressing Weaknesses
>
> W1 (evidence for hypotheses). We appreciate the request for additional evidence on the utility of the acquisition and exposure phases. The effects of the acquisition phase are demonstrated by the stage decomposition results (Figures 4, 6 & 8). In the revised Appendix H of the revised manuscript, we explicitly include an ablation for Retroactive PBB that isolates the contribution of the exposure phase by introducing a “stage 2 only” baseline. This confirms that exposure is necessary for the subsequent activation phase to take effect on $F_\text{unpaired}$, consistent with the hypothesis that the autoregressive training objective allows the model to internalise executable representations during symbolic exposure.
>
> We also note that the stage-wise ablation for Random Arithmetic was missing from the original submission and is now included. In addition, Appendix A (Figure 8) already reports stage ablations for Qwen on Formal Grammars, and we now include a stage-wise ablation on Random Arithmetic with Qwen to further illustrate consistent trends across model families.
>
> W2 (untuned baselines). It is correct that models show negligible pre-finetuning ability to execute the Random Arithmetic programs, since these programs are synthetic and unseen in pretraining. The goal of this experiment is precisely to test whether a model can *learn execution ability for some procedures purely from symbolic descriptions* that were not previously known. Thus, the zero-shot performance reflects the absence of prior exposure, not an artefact of the experimental setup. If the symbolic description was instead provided *in-context* at test time, pretrained models would perform better, but this differs fundamentally from the ability to execute programs from *parametrically internalised* knowledge acquired via SFT on their symbolic definitions.
>
> W3 (choices of SFT vs. RL). We thank the reviewer for highlighting this point. Empirically, we observed that SFT on demonstrations in stage 2 of Retroactive PBB underperformed RL (see Figure 2, right). Under SFT, models tended to memorise the surface forms of demonstrations rather than linking them back to the symbolic descriptions learned in stage 1. RL with verifiable rewards (via GRPO) more effectively forces the model to associate the symbolic representations with their execution, thereby “activating” executable behaviour. In contrast, SFT suffices for Proactive PBB because stage 1 jointly exposes symbolic descriptions and demonstrations, allowing the correspondence to be learned directly. We now clarify this empirical rationale and intuition in the revised manuscript.
>
> ### Answering Questions
>
> Q1. We clarify that the core Random Arithmetic experiments use 200 functions (100 paired + 100 unpaired). For the scaling analysis (Figure 9), we vary the number of paired functions used in the Proactive PBB meta-learning phase from 100 to 800 while using a fixed 200 unpaired. The upper limit of 800 reflects the total of 1000 available functions that we generated for the dataset. The updated text makes this explicit.
>
> Q2. Hyperparameters largely follow common values and are not tuned for specific tasks. Batch size was chosen based on GPU memory constraints.
>
> Q3. While we do not experiment with other programming languages, we demonstrate generalisation of PBB to Formal Grammars – a symbolic but non-programming domain – with consistent results (Figures 6 & 8). This supports that PBB extends beyond Python to other symbolic description languages.
>
> We again thank the reviewer for their thoughtful comments. We would be glad to clarify any remaining points that might help with the evaluation.

---

> > ### Author Response · Authors · 2025-12-02
> > **Summary of Discussion**
> >
> > **Key weaknesses raised:**
> >
> > 1. Insufficient evidence that the acquisition (proactive) and exposure (retroactive) phases matter across datasets and model families.
> >
> > 2. Lack of untuned or stage-isolated baselines to contextualise results.
> >
> > 3. Unclear justification for using SFT for Proactive PBB but RL for Retroactive PBB.
> >
> > **Main questions:** dataset size limits, hyperparameter sensitivity, and whether results generalise beyond Python.
> >
> > **Summary of how these were addressed:**
> >
> > The rebuttal expanded the empirical evidence requested for both the acquisition phase (Proactive PBB) and exposure phase (Retroactive PBB).
> >
> > Additional ablations were added, which confirm the findings in the paper but make the set of experiments more complete:
> >
> > - A stage-2-only baseline for Random Arithmetic.
> >
> > - Stage-wise ablations for Qwen.
> >
> > We also clarified that the zero-shot baselines already in the paper reflect using a model that was not finetuned and that the Formal Grammar results already generalise key findings beyond the domain Python programming.
> >
> > The motivation for using SFT vs RL across the two pipelines is now explicitly documented, and the hyperparameter and dataset-scale questions were addressed clearly. The manuscript has been updated accordingly, including explicit clarification of the 800-program scaling experiment.

---

### Official Review · Reviewer_aFMH · 2025-11-04

**Soundness:** 3
**Presentation:** 3
**Contribution:** 3
**Rating:** 6
**Confidence:** 3

**Summary:**

The paper describes how an LLM can be finetuned on a set of functions, with some of those functions having input/output examples, and it will learn to be able to compute input/output mappings for the functions that don't have input/output examples, when those functions are tested with inputs at test time they will execute correctly.
Essentially it memorizes all the functions during finetuning, and the finetuning on the input/output examples "teaches" the LLM how execute the functions so that the functions that don't have input/output pairs get correctly executed at test time on inputs.

The paper also show this finetuning process works on some other domains.

**Strengths:**

Honestly I was a bit surprised this approach worked at all when first reading the paper, I sort of assume test time access to the functions that had not been "executed" with input/output pairs at train/finetune time would be required, and that really this wouldn't work well outside of chain of thought sort of step by step walking through the functions line by line to compute the results at test time.  That functions presented at finetune time are memorized in an executable way is not what I would have expected.

Your approach is novel and unique and surprising that it works (to me) where how I would approach this problem is different (described below in weakness)

**Weaknesses:**

It feels a bit almost accidental that the way the LLM happens to encode the functions it has seen at finetune time without input/output pairs are able to "lean on" / "borrow" from the input/output pair experience of the functions that had input/output examples. It feels sort of hackey, the empirical results do show this transfer works, but it doesn't feel reliable to me.

I was unclear what the RL approach was from the paper, the SFT approach I think is more obvious what you would do at finetune time, but the RL approach is not so obvious, and I didn't see the code for it in my very brief search of the provided code.

I guess if I was to approach this problem, I would have done a COT sort of approach where at test time I would train the LLM to reproduce the complete function it's trying to execute in a thinking block, along with a chain of thought sort of scratchpad computation of interim results to execute the function, and then to close the thinking block and output the answer. I feel like that would give a more reliable chance of correctly executing the functions it saw with input/output pairs, and the functions that didn't have input/output pairs, just the LLM needs to remember the function and reproduce the function in a thinking block and execute it in a chain of thought-ish way.

But I guess your approach is novel and unique and surprising that it works (to me) where what I describe is maybe not so novel.

**Questions:**

Can you describe in more detail how the RL training was done?

---

> ### Author Response · Authors · 2025-11-14
> **Author rebuttal**
>
> We thank the reviewer for their insightful comments and for their recognition of the novel aspects of our work. We are glad they found our core finding (that LLMs can internalise executable procedures from symbolic descriptions alone) to be surprising and noteworthy.
> We would like to clarify the RL training process, as requested, and also touch on the reviewer's insightful suggestions regarding CoT execution.
> ### Clarification on the RL training process
> We thank the reviewer for pointing out the RL process was not explained clearly enough in our submitted draft. The RL training is applied only during Stage 2 of the Retroactive PBB pipeline, as we explain in Section 5.1 (line 394).
>
> Summarising, the process is the following:
> - **Stage 1:** The base model is first finetuned via SFT on the *full set of symbolic descriptions* (e.g., all Python function definitions) without any corresponding input-output (I/O) examples. The objective of this stage is for the model to memorise functions.
> - **Stage 2:** The model from Stage 1 is then finetuned on the *execution examples* (the I/O pairs) with RL. This stage teaches the model to treat the descriptions from Stage 1 as executable. RL is applied only on the paired subset of procedures.
> For this Stage 2 step, we compared standard SFT with RL. As noted in the paper (Figure 2, right, and Appendix D), the online RL algorithm GRPO was substantially more effective at eliciting the PBB capability than SFT.
>
> The specific RL training setup was as follows:
> - **Task:** The model was prompted with an execution task from the set of functions we finetuned on (e.g., "Calculate foo(10)?").
> - **Generation:** The model's task was to generate the final answer, enclosed in a specific predefined format (e.g., <<20>>). The model was free to use CoT reasoning before producing the final formatted answer.
> - **Reward Scheme:** A sparse reward was provided based only on the final generated output – this is now stated explicitly in Section 4.2 of the revision:
>   - **Correct Answer:** A positive reward (+1) was given if the final answer was numerically correct and properly formatted.
>   - **Incorrect Answer:** A neutral reward (0) was given if the answer was well-formatted but numerically incorrect.
>   - **Formatting Error:** A negative reward (-1) was given if the model failed to produce the answer in the required format.
> ### On the reliability of results
> We appreciate the reviewer's intuition that this capability might not feel reliable and agree it’s currently a nascent ability (as we also point out in the submission), however, in our experiments we do show the method consistently works across different domains, multiple model classes, and inference modes, while also demonstrating practical ways of improving the ability (the data scaling experiments in Appendix B) and discussing ways to further improve it in future research.
> ### On CoT vs Direct
> Our findings strongly support the reviewer's intuition about the power of CoT. While we show that implicit execution is possible, our results consistently demonstrate that performance is significantly improved by using CoT reasoning at test time (as shown in Figure 2, left, and Figure 3). This suggests that PBB internalises the procedure, which can then be used as a guide for more robust, explicit step-by-step computation, aligning well with the reviewer's proposed mechanism.
>
> While we don’t force CoT reasoning during RL training, we don’t restrict it either, and the model could effectively learn to recall the full function in context before returning the final answer. We have observed that this does sometimes happen.
>
> Our goal, however, was to also investigate to what extent the model could internalise the procedure so that it could execute it **implicitly** in a forward pass (what we call 'Direct' execution), without needing to "unroll" the code at test time.
> We hope to have clarified some of the reviewer’s doubts and thank the reviewer again for their valuable feedback.

---

> > ### Author Response · Authors · 2025-12-02
> > **Summary of Discussion**
> >
> > **Key weaknesses raised:**
> >
> > 1. The mechanism underlying PBB felt “accidental,” raising reliability concerns.
> >
> > 2. The RL setup was unclear in the original submission.
> >
> > 3. A suggestion that a CoT-based approach might be more reliable, and a question of how the two approaches relate.
> >
> > **Main question:** how exactly was RL carried out?
> >
> > **Summary of how these were addressed:**
> >
> > The rebuttal provided a full, precise description of the RL setup (reward scheme, formatting constraints), which was missing in the original submission.
> >
> > It also clarified reliability concerns by highlighting consistent cross-domain patterns and the scaling behaviour observed across model sizes and dataset sizes.
> >
> > The rebuttal further explained the relationship between our use of CoT and the reviewer’s CoT reasoning suggestion, and clarified why studying direct execution reveals an additional, surprising generalisation capability.
> >
> > The revised manuscript now contains all of the technical details that were requested.

---

### Official Review · Reviewer_fuDV · 2025-11-08

[review text omitted: it was posted to a different submission]

---

> ### Author Response · Authors · 2025-11-12
> **Review is for the wrong paper**
>
> Dear Reviewer,
>
> The current review is for an entirely different paper. Please could you provide a review for our submission, Programming by Backprop: Learning Behaviour from Symbolic Descriptions.
>
> Many thanks,
> The Authors

---

> > ### Comment · Area_Chair_TidP · 2025-11-12
> >
> > Noted from the AC, and yes please update the reviews if an incorrect version has been uploaded. Many thanks.

---

> > > ### Comment · Reviewer_fuDV · 2025-11-13
> > >
> > > I have already updated the review again.

---

> > ### Comment · Reviewer_fuDV · 2025-11-13
> >
> > Hi, sooo sorry about that. My sincere apologies — I accidentally pasted comments from another RDMA paper earlier. There are so many papers this year. I’ve now updated the review. Sorry again.

---

> ### Author Response · Authors · 2025-11-14
> **Author rebuttal**
>
> We thank the reviewer for their careful reading and for the constructive, detailed assessment. We are encouraged by the recognition of our contribution’s clarity, breadth, and nuanced findings. Below, we address the raised weaknesses and the main question.
>
> ### Addressing Weaknesses
>
> W1 ("limited scale and reliability"). Our goal is not to claim immediate practical deployment, but to characterise and diagnose *how* and *when* LLMs can learn behaviours from symbolic descriptions. Our experiments demonstrate that (i) current models do not reliably exhibit PBB by default, but (ii) the capability *can* be elicited via targeted finetuning, with clear positive scaling trends. Appendix B provides explicit data-scaling analyses: increasing both model size and the amount of paired meta-learning data improves performance on unpaired procedures. We view these results as identifying a promising direction: introducing symbolic descriptions earlier in training, or in larger quantities, may improve reliability – Llama-3.1-8B-Instruct reaches > 60% accuracy when executing unpaired procedures when 800 functions are used for $F_\text{paired}$ (Figure 9).
>
> W2 ("dataset scale and external validity"). We agree that synthetic tasks are not fully representative of open-domain behaviours; however, they allow us to rigorously control the structure of procedures and test how PBB generalises across paired/unpaired universes. To demonstrate external validity, we complement these with two substantially more practical domains (Leetcode and OOD ciphers), which already constitute stronger external tests than many studies on algorithmic reasoning. Appendix B shows that PBB continues to improve as we increase the number of paired functions, demonstrating that results are not tied to a specific 100/100 split. We agree that expanding to further practical domains is an important next step, but note that controlled studies of generalisation in naturally occurring data are challenging.
>
> W3 ("natural language underperforms"). This is an important point, and one we emphasise in the paper. We believe that identifying this finding is an important contribution. Algorithmic and formal tasks benefit greatly from code’s structured syntax. Code is also a substantial component of pretraining corpora, plausibly enabling models to learn procedural abstractions that transfer. Additionally, programmatic generation of specifications (as in Random Arithmetic) makes code a practical medium for constructing large-scale synthetic PBB datasets. While natural language currently performs worse, it improves with model scale. We acknowledge that our results motivate future work on representation-learning pipelines (e.g., NL $\to$ pseudocode $\to$ executable abstraction), which could systematically bridge the gap and have included discussion of this in the updated manuscript.
>
> ### Answering Questions
>
> The reviewer asks whether the reliance on structured symbolic inputs limits the practicality and scalability of PBB, given that real-world procedural knowledge can also be expressed in natural language. Our position is that PBB is a general learning paradigm, but current models exhibit a strong dependence on the *structure* of the description. Code and grammars provide explicit, unambiguous abstractions that models can internalise more reliably. Natural language lacks this regular structure and therefore poses a more challenging learning signal for PBB. However, we see this as a *scaling and representation* issue rather than a fundamental limitation:
>
> - **Positive scale-dependence:** NL PBB improves with model size, suggesting that future models may be substantially more capable in this regime.
> - **Intermediate formalisms:** Many practical workflows already translate NL specifications to structured representations (pseudocode, planning languages). PBB could be applied at these intermediate stages; our work provides evidence that the formal end of this spectrum works well.
> - **Synthetic datasets at scale:** Because code-like descriptions are easy to generate programmatically, PBB could be elicited in earlier LLM training stages, potentially allowing models to internalise general symbolic-interpretation skills before encountering downstream NL descriptions.
> - **General mechanism:** Our CFG experiments demonstrate that PBB extends outside programming entirely, to abstract formal systems. This indicates that what matters is symbolic structure, not code specifically.
>
> Thus, while NL PBB is currently weaker, the paradigm itself is not restricted to code, and our results point towards concrete avenues for future work on making NL PBB practical. We have incorporated this discussion into Appendix F of the revised manuscript.
>
> We appreciate the reviewer’s thoughtful feedback and believe the revisions will make the contributions and limitations clearer. If there are additional aspects of the work that would benefit from further clarification, we would be glad to address them.

---

> > ### Author Response · Authors · 2025-11-29
> >
> > The review reverted to the one posted by mistake. Could the AC fix it back to the correct version?
> >
> > Many thanks, The Authors

---

> > > ### Author Response · Authors · 2025-12-02
> > > **Summary of Discussion (for the corrected review)**
> > >
> > > **Key weaknesses raised:**
> > >
> > > 1. PBB is not yet reliable enough, with performance degrading as procedural complexity increases.
> > >
> > > 2. The evaluation uses primarily synthetic or controlled datasets, raising concerns about external validity.
> > >
> > > 3. Strong dependence on structured symbolic data, with natural language performing considerably worse.
> > >
> > > **Main question:** whether this limits PBB’s scalability or practicality.
> > >
> > > **Summary of how these were addressed:**
> > >
> > > The rebuttal clarified how the paper addresses these points.
> > >
> > > 1.  The work does not claim practical deployment, but analyses when and how PBB emerges, with explicit data-scaling results showing reliability improves with model and dataset size.
> > >
> > > 2. The experimental design combines controlled synthetic tasks with more realistic domains (Leetcode, OOD ciphers), and the rebuttal expanded on the external validity of this combination while acknowledging that broader domains are a natural next step.
> > >
> > > 3. The substantial dependence on structured descriptions is discussed in depth; the revised manuscript adds explicit pathways for bridging the NL-code gap via intermediate formalisms and scaling trends.
> > >
> > > The main question regarding scalability beyond symbolic representations is addressed by positioning this limitation as a current model capability issue rather than a limitation of the paradigm itself.

---

### Author Response · Authors · 2025-11-23
**Discussion Reminder**

We kindly encourage reviewers to review the rebuttals, which directly respond to all questions and weaknesses identified. We appreciate your careful consideration.

---

### Author Response · Authors · 2025-12-02
**Discussion Summaries**

We have added a short summary comment under each official review, consolidating how the rebuttal and revisions address the key weaknesses raised. These comments highlight (i) where additional ablations or clarifications were incorporated into the revised manuscript, (ii) how methodological or soundness concerns were resolved, and (iii) how the significance and novelty of the contribution are positioned relative to prior work. We hope these summaries assist the area chair in evaluating the submission alongside the full discussion.

---

### Meta-Review · Area_Chair_aFud · 2025-12-21

**Summary:**

The major concern is related to the validity of the experiments. This is expressed by the most reactive reviewer.

**Reviewer Concerns:**

As discussed by the authors, the weaknesses raised by the reviewer _pmd4_ are very relevant: (1) the perceived mismatch between the stated hypothesis and the methodology; (2) Insufficient engagement with prior program-execution and neural-interpreter literature. (3) Lack of experiments without demonstrations (direct symbolic-only setup). (4) Insufficient description of the RL setup and concerns that RL may undermine the hypothesis. (5) Lack of controlled tests isolating CoT and RL effects. [Verbatim from authors]

**Reviewer Scores:**

Reviewer pmd4's concerns have been partially solved.

---

### Decision · Program_Chairs · 2026-01-26

Accept (Poster)